# A Review on Differential Abundance Analysis Methods for Mass Spectrometry-Based Metabolomic Data

**DOI:** 10.3390/metabo12040305

**Published:** 2022-03-30

**Authors:** Zhengyan Huang, Chi Wang

**Affiliations:** 1Everest Clinical Research Corporation, Little Falls, NJ 07424, USA; 2Markey Cancer Center, Department of Internal Medicine, University of Kentucky, Lexington, KY 40536, USA

**Keywords:** differential abundance, mass spectrometry, metabolomics, zero-inflated data

## Abstract

This review presents an overview of the statistical methods on differential abundance (DA) analysis for mass spectrometry (MS)-based metabolomic data. MS has been widely used for metabolomic abundance profiling in biological samples. The high-throughput data produced by MS often contain a large fraction of zero values caused by the absence of certain metabolites and the technical detection limits of MS. Various statistical methods have been developed to characterize the zero-inflated metabolomic data and perform DA analysis, ranging from simple tests to more complex models including parametric, semi-parametric, and non-parametric approaches. In this article, we discuss and compare DA analysis methods regarding their assumptions and statistical modeling techniques.

## 1. Introduction

Metabolomics has become a mature science, with over 20 years since it was first coined in 1998 [1,2,3]. It is the study of small molecules, known as metabolites, of chemical reactions within a biological system, which directly reflects the biochemical activity and provides insights into the underlying status of the system [4]. As a key component of the omics cascade, metabolomics best represents the molecular phenotype [5,6].

Even though the diverse nature of metabolites remains a challenge in compound identification and reliable quantification, metabolomics is routinely applied to multiple disciplines in life science with the advances in Mass Spectrometry (MS) [7]. Together with its various techniques, MS has high sensitivity, high mass resolution and accuracy, and the capability to detect and quantify numerous metabolites simultaneously [7,8,9]. The common applications of MS-based metabolomics include but are not limited to metabolite structure elucidation [10,11,12], metabolic profiling [10,13,14,15], and metabolite identification [16,17,18,19].

Despite the advances that have been achieved, MS-based approaches still have detection limits, which can complicate metabolite identification and quantification [7,9,20]. The diversity of metabolites, including varied chemical structure, unclear scope of metabolic network, and dynamic range of abundance, can cause those detection limits [7,21]. One frequently seen characteristic of high-throughput MS-based metabolomics data is zero inflation, where the zero values are due to either the absent of the metabolites, abundance levels below the detection limits, or both. The zero values are referred to as point mass values (PMVs) and non-zero values are referred to as non-PMVs [22]. To distinguish the zero values caused by the two different reasons, PMVs are further classified as biological point mass values (BPMVs) and technical point mass values (TPMVs). BPMVs exist if metabolites are absent in the experimental sample for a biological reason, and TPMVs exist if metabolites present in the sample but the signal is below the detection limit for a technical reason [22,23].

The proportion of PMVs can be very large. Do et al. (2018) reported an overall missing rate of 19.41%, with 80.6% metabolites that had at least one PMV. Among those metabolites, about 10% had a rate of PMV over 70%. The average missing rate per observation is 19.6% [24]. In the study conducted by Faquih et al. (2020), the authors reported 58.6% metabolites had at least one PMV with an average PMV rate per observation at 38% [25]. Taylor et al. (2013) summarized the PMV rate in metabolomic, proteomic, and glycomic studies. The overall PMV rate for metabolomics data sets ranges from 14.63% to 28.53% [26,27]. In addition to the large proportion of PMVs, studies have also confirmed that MS-based omics data can be missing not at random (MNAR), which is caused by the censored values due to detection limits [26,27].

The large proportion of PMVs has a substantial impact on the downstream analysis as ignoring the PMVs can lead to biased results. In addition, the two types of PMVs are hard to separate during the experimental process due to detection limits. Appropriate statistical methods are required to characterize PMVs and distinguish BPMVs and TPMVs to ensure unbiased and efficient inference.

Another important issue for downstream statistical analysis is how to model the non-PMVs. Li et al. (2019) found that the non-PMVs of many metabolites in a metabolomic dataset were not normally distributed even after log-transformation [28]. As many parametric models require data normality assumption, this finding raises cautions about the choice of statistical models for robust analysis.

A major type of downstream statistical analysis for metabolomic data is the DA analysis, which identifies differentially abundant metabolic features between samples from different experimental groups. In this review, we focus on statistical methods for DA analysis and discuss the pros and cons of each method regarding their assumptions and statistical modeling techniques.

## 2. Statistical Methods for DA Analysis

Naïve approaches for DA analysis include ignoring the PMVs or imputing the PMVs with non-zero values. Specifically, one approach is to delete the PMVs and apply standard methods, such as two-sample *t*-test [29] or moderated *t*-test [22,30], to the non-PMVs. However, ignoring the zero values changes the distribution of abundance level under consideration so that the results can be biased. The other approach is data imputation, which is frequently used to handle missing data including the zero-inflation issue. There are some normalization and imputation methods developed for MS data [25,26,27,31,32]. Once the zero values are imputed, the data can be analyzed using standard statistical methods such as two-sample *t*-tests. However, as we have mentioned above, due to the complex mechanisms and MNAR nature of the data, imputation methods need to be applied case by case. It is difficult to identify a suitable imputation method for a given dataset, and an inappropriate method could induce unreliable results and inferences [27,33,34].

Statistical models that can account for zero values without the need of imputation have been developed to handle different types of zero-inflated data, where zero-inflation presents not only in metabolomic studies but also in many other medical, health care, and economical studies [35,36,37,38]. Two types of zero-inflated data are frequently seen in practice; one is zero-inflated count data and the other is zero-inflated nonnegative continuous data. A recent review summarized zero-inflated count models and their applications [39]. Reviews on zero-inflated nonnegative continuous data are also available [40,41].

In this review, we focus on statistical models that have been used to handle MS-based metabolomics data. Based on the strategy of modeling PMVs and non-PMVs, these methods can be classified into three categories: one-part tests, two-part tests, and mixture models [22]. In the following sub-sections, we summarize the methods in each category. For convenience, we first introduce the following notations. Let $Y_{ij}$ be the log-transformed abundance level and $\delta_{ij}$ be the PMV indicator ($\delta_{ij}=1$ if PMV or $\delta_{ij}=0$ if non-PMV) for the jth metabolite from the ith subject, respectively, $\lambda_{j}$ be the detection limit for the jth metabolite, and $X_{i}$ be a vector of covariates for the ith subject.

### 2.1. One-Part Tests

A one-part test considers the whole distribution of metabolite data that does not separately model PMVs and non-PMVs. It uses a single test statistic that accounts for both PMVs and non-PMVs to compare a metabolite’s abundance level between experimental groups.

#### 2.1.1. Wilcoxon Rank-Sum Test

The Wilcoxon rank-sum test was first introduced by Wilcoxon in 1945 [42] for two-group comparison problems. It is often applied when the distribution of continuous measures is not normal as an alternative non-parametric option of the two-sample *t*-test. Let $n_{1}$ and $n_{2}$ be the number of subjects in groups 1 and 2, respectively. The test statistic for comparing the abundance of metabolite j between groups is
(1)$$W_{j}=\frac{\left|U_{j}-\mu_{U}\right|-0.5}{\sigma_{U}}$$
where $U_{j}=n_{1}n_{2}+n_{1}(n_{1}+1)/2-\overset{}{\underset{i\in \mathrm{Group}\;1}{\sum }}r\left(Y_{ij}\right)$, $r\left(Y_{ij}\right)$ is the rank of $Y_{ij}$ among all observations of metabolite j, $\mu_{U}=\left(n_{1}n_{2}\right)/2$ is the mean of $U_{j}$ under the null hypothesis of no difference between groups, and $\sigma_{U}=\sqrt{n_{1}n_{2}\left(n_{1}+n_{2}+1\right)/12}$ is the standard deviation. For MS-based metabolomics data, since there are tied ranks largely due to PMVs, $\sigma_{U}$ needs to be adjusted as follows:(2)$$\sigma_{U}^{\prime }=\sqrt{\frac{n_{1}n_{2}\left(n_{1}+n_{2}+1\right)}{12}-\frac{n_{1}n_{2}\sum_{k=1}^{K_{j}}\left(t_{kj}^{3}-t_{kj}\right)}{12\left(n_{1}+n_{2}\right)\left(n_{1}+n_{2}-1\right)}}$$
where $K_{j}$ is the total number of unique ranks and $t_{kj}$ is number of ties for the kth rank for the jth metabolite.

#### 2.1.2. Truncated Wilcoxon-Test

The truncated Wilcoxon-test was proposed by Hallstrom in 2010 to handle zero-inflated data for two group comparison with equal sample size [43]. The Wilcoxon rank-sum test is performed after an equal and maximal amount of zeros are removed from each group to gain power. The method was extended to data with unequal sample size by Wang et al. (2021) [44]. Assuming the equal and maximal amount of zero observations are removed from each group,$\;n_{1}^{\prime }$ and $n_{2}^{\prime }$ observations are left. The test statistic is calculated using equations in Section 2.1.1 with $n_{2}$ and $n_{2}$ to be replaced by$\;n_{1}^{\prime }$ and $n_{2}^{\prime }$. 

#### 2.1.3. Tobit-Model

The Tobit-model [22] assumes PMVs are TPMVs caused by left censoring at the detection limit. It models data by a left-censored normal distribution. The log likelihood function for metabolite j is:(3)$$\log L\left(\mu_{j},\sigma_{j}\right)=\underset{i:\;\delta_{ij}=0\;}{\sum }\log\left\{\frac{1}{2\pi \sigma_{j}}\varphi \left(\frac{Y_{ij}-\mu_{ij}\;}{\sigma_{j}}\right)\right\}+\underset{i:\;\delta_{ij}=1}{\sum }\log\left\{\Phi \left(\frac{\lambda_{j}-\mu_{ij}}{\sigma_{j}}\right)\right\}$$
where $\mu_{ij}=\beta_{0j}+I\left(i\in \mathrm{Group}\;2\right)\beta_{1j}$, $\sigma_{j}$ is the standard deviation, and φ() and Φ() are density and cumulative distribution functions of the standard normal distribution, respectively. A likelihood ratio test is applied to test the hypothesis of $\beta_{1j}=0$ for DA analysis.

### 2.2. Two-Part Tests

A two-part test first uses two independent test statistics, one for assessing the difference in non-PMVs and the other for assessing the difference in PMVs, and then combines the two test statistics to determine the overall difference between experimental groups [22,45]. A two-part test explicitly compares the proportion of PMVs between groups, although it does not further separate PMVs into BPMVs and TPMVs.

#### 2.2.1. Two-Part t-Test

For PMVs, a Pearson’s Chi-square test statistic is applied to compare the zero proportion between the two groups. For non-PMVs, a *t*-test is applied on non-zero values to get the test statistic. The test statistics for PMVs and non-PMVs both follow the chi-square distribution with 1 degree of freedom (d.f.). Assuming the proportion of PMVs is not 0 and not 1 in both groups, the pooled test statistic, the Pearson’s Chi-square test statistic plus the square of the *t*-test statistic, follows a chi-square distribution with 2 d.f.s [22].

#### 2.2.2. Two-Part Wilcoxon Test

The two-part Wilcoxon test is constructed similarly to the two-part *t*-test, except that it uses a Wilcoxon rank-sum test instead of a *t*-test for non-PMVs [22].

#### 2.2.3. SDA

Li et al. (2019) [28] proposed a semi-parametric approach named semi-parametric differential abundance analysis (SDA), which applies a logistic regression for the PMVs (Equation (4)) and a semi-parametric model (Equation (5)) for the non-PMVs:(4)$$log\left(\frac{\pi_{ij}}{1-\pi_{ij}}\right)=\gamma_{0j}+\gamma_{j}X_{i},$$
(5)$$Y_{ij}=\beta_{j}X_{i}+\varepsilon_{ij},$$
where $\gamma_{j}$ and $\beta_{j}$ are the covariates’ effects for jth metabolite for the PMVs and non-PMVs, respectively. In Equation (5), the distribution of the independent error term $\varepsilon_{ij}$ is unspecified, which allows the metabolite abundance level to be arbitrarily distributed that can deviate from the normal distribution. SDA considers the following kernel-smoothed likelihood for parameter estimation:(6)$$L\left(\beta_{j},\gamma_{j},\gamma_{0j}\right)=\overset{N}{\underset{i=1}{\prod }}{\left[\frac{\exp\left(\gamma_{0j}+\gamma_{j}X_{i}\right)}{1+\exp\left(\gamma_{0j}+\gamma_{j}X_{i}\right)}\right]}^{Ι\left(\delta_{ij}=1\right)}\times {\left[\frac{\frac{1}{Nh}\sum_{i^{*}=1}^{N}K\left\{(Y_{i^{*}j}-\beta_{j}X_{i^{*}}-\frac{\left(Y_{i^{*}j}-\beta_{j}X_{i}\right)}{h}\right\}}{\log(Y_{ij})\left\{1+\exp\left(\gamma_{0j}+\gamma_{j}X_{i}\right)\right\}}\right]}^{Ι\left(\delta_{ij\;}=0\right)}$$
where $1/Nh\overset{N\;}{\underset{i^{*}=1}{\sum }}\left\{(Y_{i^{*}j}-\beta_{j}X_{i^{*}}-\left(\left(Y_{ij}-\beta_{j}X_{i}\right)\right)/h\right\}$ is the kernel density estimator with K(.) as a one dimensional kernel function, h as the bandwidth, and N as the sample size. For DA analysis on the effect of a covariate, SDA assesses whether the corresponding model coefficients in $\gamma_{j}$ and $\beta_{j}$ are equal to zero based on a likelihood ratio test.

### 2.3. Mixture Models

The mixture model considers PMVs as a mixture of BPMVs and TPMVs, where the TPMVs component is quantified by the left censoring probability from a parametric model on non-BPMVs (including both TPMVs and non-PMVs). As the mixture model clearly separates BPMVs and TPMVs, it provides sufficient flexibility for comparing the proportion of BPMVs, proportion of TPMVs, and mean of non-BPMVs between groups, although a parametric model assumption is required to characterize the distribution of non-BPMVs.

#### 2.3.1. Left-Inflated Mixture Likelihood Ratio Test (LIM-LRT) 

The left-inflated mixture model (LIM) combines a Bernoulli distribution and a left-censored normal distribution. It has been applied to many studies including omics [22,26,46,47,48,49]. Specifically, the distribution of abundance of metabolite j for subject i from group g (g = 1 or 2) has the following density function: $$f(Y_{ij}|p_{jg},\mu_{jg},\sigma_{jg})=\{\begin{array}{cc} p_{jg}+\left(1-p_{jg}\right)\Phi \left(\frac{\lambda_{j}-\mu_{jg}}{\sigma_{jg}^{2}}\right),\;if\;\delta_{ij}=1 & \mathrm{(7)}\\ \left(1-p_{jg}\right)\varphi \left(\frac{Y_{ij}-\mu_{jg}}{\sigma_{jg}^{2}}\right),\;if\;\delta_{ij}=0 & \mathrm{(8)} \end{array}$$where $\mu_{jg}$ is the mean, $\sigma_{jg}$ is the standard deviation, and $p_{jg}$ and $\left(1-p_{jg})\Phi (\lambda_{j}|\mu_{jg},\sigma_{jg}\right)$ are the proportions of BPMVs and TPMVs, respectively, for metabolite j from group g. Based on Equation (8), non-PMVs follow a truncated normal distribution:(9)$$f\left(Y_{ij}|\delta_{ij}=0,\mu_{jg},\sigma_{jg}\right)=\frac{\varphi \left(\left(Y_{ij}-\mu_{jg}\right)/\sigma_{jg}^{2}\right)}{\sigma_{jg}\left(1-\Phi \left(\left(\lambda_{jg}-\mu_{jg}\right)/\sigma_{jg}^{2}\right)\right)}$$

A likelihood ratio test (LIM-LRT) for the hypothesis of $\mu_{j1}=\mu_{j2}$ and $p_{j1}=p_{j2}$ is used to assess whether metabolite j is differentially abundant between groups.

#### 2.3.2. DASEV

Huang et al. (2020) noticed that the variance estimation from LIM could be unstable in presence of a large proportion of zero values, which affected the DA analysis results [50]. To address this issue, they adapted the variance shrinkage approach proposed by Smyth (2004) for microarray data to the mixture model setting, where data from the ensemble of metabolites were borrowed to achieve a more robust variance estimation of each individual metabolite [30]. Specifically, the variances of all metabolites, $\sigma_{j}^{2}$’s, are assumed to have the following common prior distribution: (10)$$\sigma_{j}^{2}\;\sim \;\mathrm{Inv}-\mathrm{Gamma}\left(\frac{d_{0}}{2},\frac{d_{0}s_{o}^{2}}{2}\right),$$
where $d_{0}/2$ and $d_{0}s_{o}^{2}/2$ are the shape and scale parameters for the inverse-gamma distribution, respectively. The $d_{0}$ and $s_{0}$ are specified as follows:(11)$$d_{0}=2m^{2}/\upsilon +4,$$
(12)$$s_{0}=\sqrt{m\left(d_{0}-2\right)/d_{0}},$$
where m and υ are the sample mean and variance for the initial estimate of $\sigma_{j}^{2}$ across all metabolites. After the shape and scale parameters are determined, iterations are done until convergence to obtain estimates of ${\hat{p}}_{jg}$ and ${\hat{\mu }}_{jg}$ by maximizing the likelihood:(13)$$L\left(p_{j1},p_{j2},\mu_{j1},\mu_{j2}|\sigma_{j}\right)=\overset{2}{\underset{g=1}{\prod }}\left\{\overset{}{\underset{i\in \mathrm{Group}\;g}{\prod }}f\left(Y_{ij}|p_{jg},\mu_{jg},\sigma_{j}\right)\right\},$$
and ${\hat{\sigma }}_{j}^{2}$ by maximizing the posterior: (14)$$p\left(\sigma_{j}^{2}|Data\right)\propto L\left(p_{j1},p_{j2},\mu_{j1},\mu_{j2}|\sigma_{j}\right){\left(\frac{d_{0}s_{o}^{2}}{2}\right)}^{\frac{d_{0}}{2}}\frac{\sigma_{j}^{2\left(-1-\frac{d_{0}}{2}\right)}}{\Gamma \left(\frac{d_{0}}{2}\right)}\exp\left(-\frac{d_{0}s_{o}^{2}}{2\sigma_{j}^{2}}\right).$$

After all model estimates are obtained until convergence, a likelihood ratio test is applied for DA analysis. Huang et al. (2020) also extended LIM to allow covariate adjustment, where a logistic regression model was used to characterize the association between covariates and the proportion of BPMVs and a linear model was used to characterize the association between covariates and the mean of non-BPMVs [50].

### 2.4. Model Comparison

Simulation studies that compared the performance of different methods were conducted in the literature. Gleiss et. al. compared models including Wilcoxon rank-sum test, truncated Wilcoxon test, Tobit-model, two-part *t*-test, two-part Wilcoxon test, and LIM-LRT, Huang et al. compared LIM-LRT and DASEV, and Li et al. compared two-part *t*-test, two-part Wilcoxon test, and SDA [22,28,50]. In summary, if the proportion of BPMVs are similar between the two groups, one-part tests generate acceptable results. Two-part tests have more reliable estimates comparing to one-part tests especially when the TPMVs proportions are not too high [22]. Two-part Wilcoxon test shows good performance if TPMVs can be ruled out [22]. SDA is able to handle both normally and non-normally distributed features simultaneously, and outperforms two-part *t*-test and two-part Wilcoxon test for non-normally distributed features [28]. Mixture models, especially DASEV, can provide less biased estimates on both proportions of TPMVs and BPMVs when the TPMV proportion is high [22,50]. LIM-LRT, DASEV, and SDA all yield good true positive rates when the PMV proportion is not very high [22,28,50].

## 3. Practical Guidelines

Table 1 summarizes the modeling technique and assumption for each DA analysis method. In practice, the choice of appropriate methods to use depends on the characteristics of the specific dataset. The following factors need to be considered.

The composition of PMVs. As different methods model PMVs in different ways, we would suggest first investigating the composition of PMVs before performing DA analysis. One can draw a histogram to investigate the empirical distribution of abundance level. If the observed PMV proportion is substantially higher than the extrapolation of the distribution of non-PMVs, it would indicate the presence of BPMVs. Under such situation, the Tobit-model, which assumes PMVs are all from TPMVs, may not be appropriate. Further, if one wants to separate the proportions of BPMVs and TPMVs, the mixture model-based approaches, LIM-LRT and DASEV, would be preferred.

Data normality. We would also suggest checking data normality by using the Q-Q plot, Kolmogorov-Smirnov test, and Shapiro-Wilk test. If the data substantially deviate from normal distributions, non-parametric and semi-parametric methods that do not require the normal assumption would be preferred. Those methods include Wilcoxon rank-sum test, truncated Wilcoxon test, two-part Wilcoxon test, and SDA.

Sample size. Although non-parametric and semi-parametric methods are robust to distributional assumptions, they typically require larger sample sizes compared to parametric methods. For example, the Wilcoxon rank sum test requires a sample size of at least 16 [51,52]. Therefore, if the experiment only has a few replicates per treatment group, using a parametric method is more feasible.

Confounder adjustment. Adjusting for confounders, e.g., age and sex, is allowed for some parametric and semi-parametric methods such as DASEV and SDA. Therefore, for studies with a complex design and/or presence of confounders, those methods would be preferred.

Finally, it is always a good practice to consider more than one method and compare the results to make more robust inference. 

## 4. Discussion

Handling zero inflation is an important task for analyzing MS-based metabolomic data. The characteristics of zero-inflated data need to be carefully assessed in order to choose appropriate statistical methods for data analysis, which will impact analysis results and interpretation. In this paper, we have reviewed a variety of statistical methods to model zero-inflated data for DA analysis. By comparing these methods in the aspects of assumptions and statistical modeling techniques, we have provided guidelines for choosing appropriate methods in practical situations. Our review focuses on cross-sectional studies. For the more complex longitudinal metabolomics studies on the progression of diseases [53,54,55], current approaches consider mixed effect models [56,57,58]. New method developments to handle the zero-inflation issue are highly desired to achieve more robust performance and increase the predictability of such studies. 

In addition to DA analysis, the zero inflation issue also broadly affects many other types of downstream analysis of metabolomic data such as cluster analysis [59], disease diagnostic modeling [60], pathway analysis [61,62,63], and multi-omics analysis [64]. For example, a common approach for pathway analysis is the overrepresentation analysis [61,63], which identifies enrichment of a metabolic pathway by assessing the overrepresentation of metabolites from the pathway in a list of metabolites of interest compared to the background. The overrepresentation analysis is based on an input of a list of metabolites of interest, which is usually the list of differentially abundant metabolites from a DA analysis. Thus, the strategy of handling PMVs in DA analysis will have an impact on the results of pathway analysis. 

## Figures and Tables

**Table 1 metabolites-12-00305-t001:** Comparison of statistical methods for DA analysis.

Category	Method	Able to Distinguish TPMVs and BPMVs	Free of Data Normality Assumption	Available R Function/Package	References
One-part test	Wilcoxon rank-sum test	N	Y	wilcox.test	[42]
	Truncated Wilcoxon test	N	Y	https://rdrr.io/github/chvlyl/ZIR/	[43,44]
	Tobit-model	N	N	VGAM (https://cran.r-project.org/web/packages/VGAM/index.html)	[22]
Two-part test	Two-part *t*-test	N	N	t.test binom.test	[22]
	Two-part Wilcoxon test	N	Y	wilcox.test binom.test	[22]
	SDA	N	Y	SDAMS (https://bioconductor.org/packages/release/bioc/html/SDAMS.html)	[28]
Mixture Model	LIM-LRT	Y	N	https://cemsiis.meduniwien.ac.at/en/kb/science-research/software/statistical-software/limlrt/	[22,26,46,47]
	DASEV	Y	N	http://sweb.uky.edu/~cwa236/DASEV.html	[50]

Y: Yes; N: No. All the hyperlinks were accessed on 25 March 2022.

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
