# Peer review of "A Review on Differential Abundance Analysis Methods for Mass Spectrometry-Based Metabolomic Data"

_metabolites, 2022, doi:10.3390/metabo12040305_

Round 1

Reviewer 1 Report

Huang and Wang present a review manuscript on statistical inference methods for handling zero-inflated metabolomics data. Although the content of the manuscript has a good deal of overlap to for example Gleiss et al (2015), I believe that having a review article with a focus on metabolomics is beneficial for the community. 

I recommend having a short section on multivariate zero-inflated models to serve as a starting point for researchers and guide them to other resources. 

In Table 1, It would be beneficial to add a column mentioning possible implementation of these algorithms in for example R, Python or Matlab. That would be beneficial for practitioners who are both after theoretical part and also usage of these methods.

I did not see much discussion about the power of these methods. Much of the discussion was centred around the assumptions and method selection. It would be great to mention or refer to simulation studies where some of these methods have been compared. 

Although I wanted to suggest running a simulation and comparing the methods. But I believe that goes beyond the scope of a review article. So I can only suggest it if the authors are interested.

Author Response

Response to Reviewer 1 Comments

Huang and Wang present a review manuscript on statistical inference methods for handling zero-inflated metabolomics data. Although the content of the manuscript has a good deal of overlap to for example Gleiss et al (2015), I believe that having a review article with a focus on metabolomics is beneficial for the community. 

Point 1: I recommend having a short section on multivariate zero-inflated models to serve as a starting point for researchers and guide them to other resources. 

Response 1: We added a paragraph (lines 84-90) to describe zero-inflated models in general and reference available resources.

Point 2: In Table 1, It would be beneficial to add a column mentioning possible implementation of these algorithms in for example R, Python or Matlab. That would be beneficial for practitioners who are both after theoretical part and also usage of these methods.

Response 2: We added a column “Available R Function / Package” to Table 1 to list R functions or packages for the statistical methods.

Point 3: I did not see much discussion about the power of these methods. Much of the discussion was centered around the assumptions and method selection. It would be great to mention or refer to simulation studies where some of these methods have been compared. 

Response 3: We added a section 2.4. (line 207-221) to summarize results of simulation studies from three papers that compared the performance of different methods.

Point 4: Although I wanted to suggest running a simulation and comparing the methods. But I believe that goes beyond the scope of a review article. So I can only suggest it if the authors are interested.

Response 4: We greatly appreciate the comment. A comprehensive and well-designed simulation study for comparing methods under various scenarios is very valuable and would be a separate paper.

Reviewer 2 Report

It is quite obvious to me that this review would be of interest for those working in fiends such as MS for metabonomics.

I could not find any errors in the descriptions of the methods.

However, I always appreciate to see a practical application, and so I am a little frustrated not to have seen the various procedures applied to 2 or 3 data sets with the characteristics discussed in Table 1.

It would also be of interest for practicioners to have access to the corresponding functions in one of the commonly used languages.

There are a few grammatical problems, but nothing that would make the paper difficult to follow.

I suggest the authors give the full name of the SDA method

(I think it must be "semi-parametric differential abundance")

Author Response

Response to Reviewer 2 Comments

It is quite obvious to me that this review would be of interest for those working in fiends such as MS for metabonomics.

I could not find any errors in the descriptions of the methods.

Point 1: However, I always appreciate to see a practical application, and so I am a little frustrated not to have seen the various procedures applied to 2 or 3 data sets with the characteristics discussed in Table 1.

Response 1: Thank you very much for the comment. Our review focuses on providing an overview of available statistical methods as well as their assumptions and modeling techniques. A comprehensive comparison of those methods using well-designed simulation studies and real data analysis would have sufficient scientific merit to be presented as a separate paper beyond the scope of this review.

Point 2: It would also be of interest for practicioners to have access to the corresponding functions in one of the commonly used languages.

Response 2: We added a column “Available R Function / Package” to Table 1 to list R functions or packages for the statistical methods.

Point 3: There are a few grammatical problems, but nothing that would make the paper difficult to follow.

I suggest the authors give the full name of the SDA method

(I think it must be "semi-parametric differential abundance")

Response 3: We added the full name for SDA at line 155-156.

Reviewer 3 Report

This review is well organized and presented and therefore it can be published in the current form

Author Response

Response to Reviewer 3 Comments

This review is well organized and presented and therefore it can be published in the current form

Response: Thank you very much for your positive feedback.